# Optimization of Chitosan Glutaraldehyde-Crosslinked Beads for Reactive Blue 4 Anionic Dye Removal Using a Surface Response Methodology

**DOI:** 10.3390/life11020085

**Published:** 2021-01-25

**Authors:** Johanna Galan, Jorge Trilleras, Paula A. Zapata, Victoria A. Arana, Carlos David Grande-Tovar

**Affiliations:** 1Grupo de Investigación Ciencias, Educación y Tecnología—CETIC, Programa de Química, Universidad del Atlántico, Carrera 30 No 8–49, Puerto Colombia 081008, Colombia; jgalan@mail.uniatlantico.edu.co; 2Grupo de Compuestos Heterociclicos, Programa de Química, Universidad del Atlántico, Carrera 30 No 8–49, Puerto Colombia 081008, Colombia; jorgetrilleras@mail.uniatlantico.edu.co; 3Grupo de Polímeros, Facultad de Química y Biología, Universidad de Santiago de Chile, USACH, Casilla 40, Correo 33, Santiago 9170020, Chile; paula.zapata@usach.cl; 4Grupo de Investigación de Fotoquímica y Fotobiología, Programa de Química, Universidad del Atlántico, Carrera 30 No 8–49, Puerto Colombia 081008, Colombia

**Keywords:** adsorption, reactive blue 4 dye, crosslinking chitosan beads, glutaraldehyde, experimental design, swelling degree, removal efficiency

## Abstract

The use of dyes at an industrial level has become problematic, since the discharge of dye effluents into water disturbs the photosynthetic activity of numerous aquatic organisms by reducing the penetration of light and oxygen, in addition to causing carcinogenic diseases and mutagenic effects in humans, as well as alterations in different ecosystems. Chitosan (CS) is suitable for removing anionic dyes since it has favorable properties, such as acquiring a positive charge and a typical macromolecular structure of polysaccharides. In this study, the optimization of CS beads crosslinked with glutaraldehyde (GA) for the adsorption of reactive blue dye 4 (RB4) in an aqueous solution was carried out. In this sense, the response surface methodology (RSM) was applied to evaluate the concentration of CS, GA, and sodium hydroxide on the swelling degree in the GA-crosslinked CS beads. In the same way, RSM was applied to optimize the adsorption process of the RB4 dye as a function of the initial pH of the solution, initial concentration of the dye, and adsorbent dose. The crosslinking reaction was investigated by scanning electron microscopy (SEM), Fourier transformed infrared spectroscopy (FTIR), and X-ray diffractometry (XRD). The design described for the swelling degree showed an R^2^ (coefficient of determination) adjusted of 0.8634 and optimized concentrations (CS 3.3% *w/v*, GA 1.7% *v/v*, and NaOH 1.3 M) that were conveniently applied with a concentration of CS at 3.0% *w/v* to decrease the viscosity and facilitate the formation of the beads. In the RB4 dye adsorption design, an adjusted R^2^ (0.8280) with good correlation was observed, where the optimized conditions were: pH = 2, adsorbent dose 0.6 g, and initial concentration of RB4 dye 5 mg/L. The kinetic behavior and the adsorption isotherm allowed us to conclude that the GA-crosslinked CS beads’ adsorption mechanism was controlled mainly by chemisorption interactions, demonstrating its applicability in systems that require the removal of contaminants with similar structures to the model presented.

## 1. Introduction

The dyes used by different industries, such as textiles, leather, paper, and plastics, among others, is of great interest, because the discharge of dye effluents into water generate persistence, bioaccumulation, and toxicity [1]. The dye’s presence in aquatic environments reduces the penetration of light and oxygen, inhibiting photosynthetic activities. Likewise, can potentially lead to carcinogenic and mutagenic diseases in aquatic species and humans [1,2]. It is estimated that, of the 800,000 tons of colorants produced annually, 20% ends up in water bodies in the final stages of industrial processes [3,4]. 

Wastewater has been shown to contain dye concentrations ranging from 10 mg/L to 200 mg/L [5]. Due to this growing environmental problem, some countries, such as the United Kingdom, have incorporated restrictions and laws to regulate the concentration of colorants present in bodies of water. However, the lack of control in handling the growing situation is still evident, especially in Latin America [6,7]. In this way, it is necessary to search for technologies that allow eliminating the colorants present in the different effluents.

Among many reported techniques, the adsorption process is commonly used due to its remarkable ability to remove dyes from aqueous solutions. However, the use of some sorbents (e.g., activated carbon) can be expensive and challenging alternatives to regenerate [8]. 

In recent years, numerous, inexpensive, and effective alternatives have been developed, including chitosan (CS), which has attracted particular attention because it is renewable, has high availability, and does not generate toxic products [9,10,11]. CS is suitable for removing anionic dyes since it has favorable properties, such as acquiring a positive charge and a typical macromolecular structure of polysaccharides [12,13]. 

However, CS easily dissolves in an acidic medium (pH < 5), which limits its use due to the high protonation of its amino groups, leading to instability and dissolution of the material in the medium [14]. Consequently, various physical and chemical modifications have been applied to its molecular structure, such as the formation of spherical beads, which increase the porosity of CS and improve its surface area [15]. Another alternative has been the addition of crosslinking agents, such as glutaraldehyde (GA), which allow better mechanical resistance and stability in a wide pH range [16]. Studies conducted by Ngah and Fatinathan [17] show how GA-crosslinked CS beads have a greater adsorption capacity to remove *p*-nitrophenol than physical modifications of CS in the form of flakes.

Likewise, the implementation of low molecular weight (LMW) CS has proven to be better biocompatible and biodegradable than high molecular weight (HMW) CS [18].

Its use at LMW provides a more excellent facility to interact with the dye because its chains are shorter and allow better access to the available adsorption sites [19]. On the other hand, although various studies have been reported on the removal of anionic dyes, there is a need to evaluate and optimize the variables that influence the processes of obtaining and adsorption of modified beads with the application of classical statistical models, a design of experiments with response surface methodology (RSM) approaches, which determine and quantify the factors that could influence the variable of interest.

Therefore, the purpose of this research was to evaluate the development of low molecular weight CS beads crosslinked with GA to maximize the removal efficiency of reactive blue dye 4 in an aqueous solution.

## 2. Materials and Methods

### 2.1. Experimental Design

A central composite experimental design (CCED) was used to evaluate the swelling degree of CS beads cross-linked with GA and the removal efficiency of the anionic dye RB4. Both designs consisted of conducting 32 randomized experiments, including duplicates. Three series of experiments were developed: (a) a factorial design (2^k^) with three factors (k = 3), each with two levels (maximum, +1 and minimum, −1); (b) axial points with coded values α = 2^k/4^ = ±1.6817; and (c) duplicate of the central point. For the first design, input factors were the concentration of CS, NaOH concentration, and GA concentration. The response factor was the swelling degree. For the second design, input factors were the adsorbent dose, initial concentration of the dye, and the dye solution’s pH. The response factor was dye removal efficiency.

The experiment design, analysis of variance (ANOVA), response surface methodology, and optimization were performed using Statgraphics Centurion XVI software (StatPoint Technologies, Inc., The Plains, USA) [20]. Second-order mathematical models were created to optimize the factors studied, according to Equation (1) [21].
(1)$$Y=\beta_{0}+\;\overset{k}{\underset{i=1}{\sum }}\beta_{i}X_{i}+\;\overset{k}{\underset{i=1}{\sum }}\beta_{ii}X_{i}^{2}+\;\underset{i=1\;<}{\sum }\underset{j=1}{\sum }\beta_{ij}X_{i}X_{j\;}+\;\varepsilon$$
where *Y* is the adjusted response variable, *i*, *j* linear, and quadratic coefficients, *β* is the regression coefficient, *k* is the number of optimized factors, and *ε* the experimental error.

### 2.2. Preparation of GA-Crosslinked CS Beads

Concentrations between 1.7 (% *w/v*) and 3.3 (% *w/v*) of CS (75–85% deacetylation degree and low molecular weight of 50–190 kDa, Merck KGaA, Darmstadt) solution were prepared in 1 (% *v/v*) acetic acid solution and homogenized with a digital disperser (ULTRATURRAX IKA T-25, Merck KGaA, Darmstadt, Germany).

The above CS solutions were added dropwise into sodium hydroxide solutions with a concentration range of 0.7 M to 2.3 M, under constant stirring, to allow the formation of CS beads with controlled spherical size. The beads formed were removed from the solution, washed with distilled water until a neutral pH value was obtained. The CS beads were immersed for 12 h in 25 mL of GA solutions with a concentration range of 1.7 (% *v/v*) to 3.3 (% *v/v*). The crosslinked beads were filtered from the solution, washed vigorously with distilled water to remove any unreacted particles, and dried in an oven at 60 °C for 10 h [22].

### 2.3. Swelling Test of Crosslinked Beads

One gram of GA-crosslinked CS beads obtained in each experiment was immersed in distilled water at room temperature and neutral pH. After 24 h, the beads were filtered, excess water was removed with filter paper, and weighed to constant weight. The swelling degree was calculated with Equation (2).
(2)$$\mathrm{Swelling}\;\mathrm{degree}\;\left(\%\right)=\;\frac{w_{1}-w_{0}}{w_{0}}\times 100$$
where *W*_0_ (g) is the dry mass and *W*_1_ (g) is the wet mass after 24 h [23]. Furthermore, the crosslinked beads’ behavior was studied at pH values of 3 and 10, adjusting with NaOH (0.1 M) and HCl (0.1 M).

### 2.4. Calibration Curve of RB4 Dye

Standard solutions were prepared at different concentrations of the RB4 dye (1, 5, 15, 25, 35, 45, 55, 60, 65, 70 mg/L). The above solutions’ absorbances were measured in the UV-VIS spectrophotometer (Mapada UV-1600, Shanghai, China), at a maximum length of 599 nm. The model was estimated using linear regression analysis to obtain the dye concentrations in the adsorption processes.

### 2.5. Adsorption Study of RB4 Dye

A stock solution of 500 mg/L of the RB4 dye was prepared. From the stock solution, solutions were prepared in a range of 5 mg/L to 55 mg/L. Each experiment’s pH was adjusted with minimum and maximum values of 2.0 and 7.0 by adding HCl (0.01 M) and NaOH (0.01 M). Next, 25 mL aliquots were taken, and their initial absorbances were measured at a maximum length of 599 nm. Subsequently, specific amounts from 0.1 g to 0.6 g, corresponding to a dose of adsorbent between 0.1 g/25 mL and 0.6 g/25mL, were added. The samples were placed in the dark with shaking at a speed of 400 rpm to prevent dye photolysis. After 48 h, following beads filtration, the absorbance of the RB4 dye in the supernatant was measured. The dye removal efficiency was determined with Equation (3).
(3)$$R\;\left(\%\right)=\frac{\left(C_{o}-C_{e}\right)}{C_{o}}\times 100$$
where *R* (%) is the removal efficiency, *C*_o_ and *C*_e_ (mg/L) are the initial concentration and the final or equilibrium concentration of the anionic dye RB4 [9].

### 2.6. Characterization of GA-Crosslinked CS Beads

Scanning electron microscopy (SEM) (JSM-6490LA, JEOL, Tokyo, Japan) and Fourier transformed infrared spectroscopy—attenuated total reflectance (FTIR-ATR) (IRAffinity-1, Shimadzu, Kyoto, Japan) were used to determine the surface morphology of the adsorbents and the presence of functional groups. The crystalline phase of the cross-linked beads was detected by X-ray diffractometry (XRD) (PANalytical X’Pert PRO, Malvern Panalytical, Malvern, United Kingdom) using radiation of Cu Kα1 (1.540598 Å) and Kα2 (1.544426 Å), in a range of 2θ comprised from 5° to 50°. Equation (4) shows the percentage of crystallinity (Xc %) of each pearl obtained from their diffractograms using the Nara-Komiya methodology [24].
(4)$$X_{c}\;\left(\%\right)=\;\frac{\left(A_{c}\right)}{\left(A_{T}\right)}\times 100$$
where A_c_ is the area of the peaks representing the crystalline regions and A_T_ is the total area of the crystalline and amorphous region.

### 2.7. Kinetic Experiment

A 500 mg/L stock solution of the RB4 dye was used to take 25 mL of a 35 mg/L solution, and its initial absorbance was measured. Then 0.4 g/25 mL of GA-crosslinked CS beads were added. This solution was stored in the dark under constant stirring at 400 rpm and room temperature. Twenty absorbance measurements were taken at different time intervals during 48 h, taking small aliquots of the stored solution. The amount of equilibrium adsorption, Q_e_ (mg/g), was calculated from Equation (5) [25]. The kinetic models described in Appendix A were analyzed using OriginPro 8 software (Origin Lab Corporation, Northampton, MA, USA).
(5)$$Q_{e}=\frac{(C_{o}-C_{e})\times V}{W}$$
where *C*_o_ and *C*_e_ (mg/L) are the liquid-phase concentrations of dye initially and at equilibrium, respectively, *V* (L) is the volume of the dye, and *W* (g) is the weight of the adsorbent.

### 2.8. Isothermal Experiment

RB4 dye solutions of 5 mg/L to 65 mg/L with a pH adjusted to 3.0 were prepared from the 500 mg/L stock solution. The isothermal experiments were carried out under the same conditions as the kinetic experiment. After 48 h of constant stirring, the beads were separated by filtration, and the final absorbances were measured. Subsequently, the adsorption equilibrium models described in Appendix A were plotted using OriginPro 8 software (Origin Lab Corporation, Northampton, MA, USA). The adsorption capacity obtained at equilibrium was determined with Equation (5).

## 3. Results

### 3.1. Optimization for the Preparation of Cross-Linked CS Beads

The optimization of the GA-crosslinked CS beads was evaluated on the swelling degree. The swelling degree accounts for the expansion of chitosan’s porous structure, which allows the interaction of the dye molecules with the adsorption sites [12]. Table 1 shows the complete experimental design of obtaining the GA-crosslinked CS beads and their respective responses to the degree of swelling obtained in each experimental run.

The contributions of the operational and interaction factors that influenced the swelling degree were evaluated by ANOVA (Table 2). The factors in order of significance and contribution at a confidence level of 95% (*p* < 0.05) were: CS concentration (A), GA concentration (B), quadratic interaction of CS concentration (AA), quadratic interaction of GA concentration (BB), interaction concentrations of CS, and GA (AB).

The model’s effectiveness was evaluated based on the coefficient of determination, R^2,^ and adjusted R^2^. R^2^ (90.74 %) values and adjusted R^2^ (86.34 %) indicated a good correlation between the input and output variables [26]. Adequate precision (14.11) indicated an adequate signal, and this model can be used to navigate the design space. Similarly, based on the experimental data, the adjusted quadratic model equation was established (Equation (6)).
(6)$$\begin{array}{cc} \mathrm{Swelling}\;\mathrm{degree}\; & \left(\%\right)\\ & =\;229.35\;-\;71.4949A\;-\;86.7575B-\;11.6602C\\ & +\;17.961A^{2}\;-\;6.2225AB\;+\;1.9125AC+\;17.9471B^{2}\\ & +\;2.7875BC-\;0.246556C^{2} \end{array}$$

A is the concentration of CS, B is the concentration of GA, and C is NaOH concentration. The Pareto chart plots the standardized effects of each factor and their interactions. The positive (+) or negative (-) effect on the response factor is given by the increase in the factor level [27]. Figure 1 represents the standardized effects for the swelling degree. The GA concentration factor was the variable that had the most significant negative influence, and the CS concentration factor was the variable that had the most significant favorable influence on the response. The increase in GA concentration produced a higher consumption of the amino groups in the crosslinking reaction. Consequently, the crosslinked CS beads had a lower ability to bind to water molecules due to reduced active sites [28,29]. 

On the other hand, the swelling degree was not affected by the NaOH concentration. This result might be due to the limited range in which this input variable was studied. A reported study indicated that when the NaOH concentration is increased in a range from 1 to 5 M, it is possible to reduce the water content in the porous beads [30]. That is, the percentage of the swelling degree decreases. Additionally, AA and BB’s significant quadratic factors indicated the presence of curvatures in the model and a possible optimal value of the swelling (%) degree for certain variables analyzed.

The response surface plot of Figure 2 shows that the swelling degree at neutral pH is favored with increasing CS concentration. The maximum predicted swelling of 60.65% was reached with a concentration of 3.3 (% *w/v*) of CS, 1.7 (% *v/v*) of GA, and 1.3 M NaOH. Similarly, the mentioned conditions were validated, obtaining a 50% swelling degree.

The GA-crosslinked CS beads response model was recommended for prediction purposes within the selected range since it translated more than 70% of the response in terms of adjusted R^2^ (0.8634). However, the surface graph shows that the most suitable treatment was found towards a point outside the experimental region, suggesting expanding the study range in the direction where RSM indicated the possible best treatment (towards the region of warm colors). It should be noted that increasing the concentration of CS implies an increase in viscosity and a more challenging bead formation. Therefore, the experimental adsorption design applied in this work was carried out at concentrations of 3.0 (% *w/v*) of CS, 1.7 (% *v/v*) of GA, and 1.3 M of NaOH, where the swelling degree obtained was 48%.

On the other hand, the cross-linked beads’ swelling was determined under the influence of acidic pH (3) and basic pH (10), reaching swelling percentages of 90% and 14%, respectively. This behavior was explained because CS is a weak base with a *pKa* of around 6.4 [31]. Therefore, at a pH value below *pKa*, the amino groups in the CS molecules ionize into ammonium ions, acting as repulsive forces between polymer chains, allowing a higher swelling degree [28].

On the other hand, at a pH above *pKa*, a more significant number of OH^−^ ions are produced in the medium, and consequently, the protonation of the amino groups is minimal, leading to a low swelling degree [28].

### 3.2. Preparation of the GA-Crosslinked CS Beads

Obtaining the GA-crosslinked CS beads required an acid-base reaction and, subsequently, cross-linking. Initially, the CS amino groups’ protonation was carried out in acetic acid solution (Figure 3a). Subsequently, a neutralization reaction took place due to the addition of a NaOH solution, which favored the formation of uniform, spherical, and porous gel beads (Figure 3b) [30]. Finally, with the crosslinking agent (GA) addition, the CS amino groups reacted with generating chemical cross-links of imines (Figure 3c). However, several studies have indicated that the CS–GA reaction can undergo self-oligomerization by aldol condensation, producing a longer chain in the crosslinking structure [32]. In this aspect, the crosslinking reaction generated an orange color that corresponded to the presence of unsaturated bonds such as C = N (imine), C = C (double bond), and C = O (aldehyde). This increase in unsaturated bonds can lead to color intensification for some beads [31,32,33,34].

### 3.3. Optimization for the Adsorption of the RB4 Dye in an Aqueous Solution

A calibration curve (Figure 4) was prepared at a λmax of 599 nm to quantify the RB4 dye experiments. The quality of the curve was evaluated based on the determination coefficient R^2^. The value of R^2^ (99.95%) indicated that the experimental points were correctly adjusted to the mathematical model of the equation.

In the adsorption processes, the GA-crosslinked CS beads were prepared from a concentration of CS at 3.0 (% *w/v*), GA at 1.7 (% *v/v*), and NaOH at 1.3 M. Table 3 shows the matrix of CCED experimental variables with RB4 removal efficiency responses.

According to Table 4, the analysis of variance showed an R^2^ and an adjusted R^2^ determination coefficient of 88.35% and 82.80%, respectively, indicating that the main effects (A, B, C) and the quadratic effects (B^2^ and C^2^) are significantly different from zero at a 95% confidence level (*p* < 0.05). Adequate precision (14.03) indicated an adequate signal, and this model can be used to navigate the design space.

Based on the experimental data, a second-order polynomial equation (Equation (7)) was used to predict the removal efficiency of the RB4 dye. The mathematical model obtained showed high predictive quality since the predicted values and the experimental values presented a good correlation.
(7)$$\begin{array}{c} \mathrm{Removal}\;\mathrm{efficiency}\;\left(\%\right)\\ =222.06+36.428A-48.3522B-2.98722C+9.67765A^{2}+8.3828AB\\ -0.27432AC+\;3.72639B^{2}-0.0861647BC\\ +\;0.0381125C^{2} \end{array}$$
where *A* (g) is the adsorbent dose, *B* is the pH, and *C* (mg/L) is the concentration of the RB4 dye.

Figure 5 shows the Pareto chart of standardized effects for RB4 dye removal with the GA-crosslinked CS beads. The pH (B) and concentration of the RB4 dye (C) were the most significant factors affecting the response variable. Both factors showed negative signs, which indicates that the percentage of removal of the dye increased with the decrease in its levels. In contrast, the adsorbent dose (A) positively affected the percentage of removal efficiency of RB4 when the values in its levels were increased. Likewise, the quadratic interactions (CC and BB) had a significantly small effect, indicating a small curvature in the model.

The solution’s pH is a variable of great importance in the dye’s removal efficiency since it can change the adsorbent’s surface charge to influence the adsorption process [35]. When the RB4 dye solution had low pH values, protonation of the amino groups occurred due to the high concentration of protons available in the solution. Protonation of the amino groups led to increased interaction capacity between negatively charged dye molecules and the GA-crosslinked CS beads’ protonated surface. In contrast, when the dye solution took on high pH values, the protonation of the available amino groups decreased, and consequently, the interaction capacity between the adsorbate and the adsorbent was reduced [36,37,38]. On the other hand, it is crucial to determine the dye species that predominate with the pH change since each species may react differently. According to reported studies [39], the RB4 dye has three predominant and deprotonated species in the pH range of 4 and 14 with estimated *pKa* values equal to 0.80, 1.44, 7.83, and 12.05. These data show that at pH 4, the two sulfonate groups are dissociated. At a pH 10 and 14, the amino groups and dissociated sulfonate groups coexist. Therefore, the acidity of the sulfonate groups in the structure of the RB4 dye maintains the anionic charges throughout the pH range, and consequently, the dye solution’s pH will affect the adsorption rate. In this sense, in the adsorption with GA-crosslinked CS beads, at a pH of 3, it could be assumed that the sulfonate groups are the species that interact with the adsorbent through an electrostatic interaction [40].

On the other hand, the adsorbent and adsorbate concentrations influenced the efficiency of the removal process. An increase in the amount of GA-crosslinked CS beads meant more significant adsorption sites on the adsorbent’s surface [41,42,43]. On the contrary, a high dye concentration generated a high electrostatic repulsion between the dye molecules and the free adsorption sites’ saturation, which caused low adsorption of the adsorbate molecules on the surface GA-crosslinked CS beads [44].

Figure 6, the 3D response surface plot for the interaction of the initial concentration of the RB4 dye and pH is illustrated, at an adsorbent dose of 0.35 g/25 mL as a central value. It was evidenced that the dye’s initial concentration influenced the saturation of the adsorption sites in the GA-crosslinked CS beads and that the pH of the solution was responsible for the surface charge of the adsorbent. Therefore, its interaction effect was influential in the response. The dye’s removal efficiency conditions were maximized: 0.6 g/25 mL of adsorbent dose, pH 2, and an initial concentration of adsorbent of 5 g/L. The maximum response value predicted by the adjusted model was 100% removal efficiency of the RB4 dye. The operating conditions mentioned were experimentally applied to confirm the optimized treatment, and it was evidenced that the dye was entirely removed.

### 3.4. Characterization of the GA-Crosslinked CS Beads

For the modified CS beads’ characterization, morphological analysis was carried out by SEM, FT-IR, and XRD.

The SEM analysis showed that the treatment of the GA-crosslinked CS beads modifies their surface texture. Figure 71a–3a shows the non-crosslinked CS beads’ interior, GA-crosslinked CS beads, and GA-crosslinked CS beads after adsorption. It was evidenced that inside the adsorbents, all had a smooth surface. This confirmed that the chemical modification and adsorption processes did not affect the internal structure of the material. The CS images on the surface Figure 71b–1d show a smooth texture, while the CS beads Figure 72b–2d show a rough and porous surface structure. This result indicated that GA had chemically modified the CS beads. In contrast, the surface of GA-crosslinked CS beads after adsorption was slightly more compact, as a product of the adsorption of molecules of the dye RB4 Figure 73b–3d [45,46].

On the other hand, the FT-IR spectrum of the CS beads without crosslinking (Figure 8A) shows characteristic broadband at 3324 cm^−1^ due to the superposition of the OH’s stretching vibrations and NH_2_ functional groups in CS. Bands at 2875, 1022, and 1370 cm^−1^ are attributed to stretching vibration C–H, C–O, C–N, respectively, indicating the polysaccharide structure of CS [31,34,47]. The band indicated at 1576 cm^−1^ is due to the bending of the primary amine’s N–H bond. A band centered at 1646 cm^−1^ is assigned to the C=O stretching vibration in the amide group’s CS due to partial deacetylation of chitin to produce CS [34]. The crosslinking reaction between GA and CS amino groups produces a stretch band around 1650 cm^−1^ that belongs to the imine bond (C=N) typical of crosslinking [34,46]. This band is observed in the spectrum of GA-crosslinked CS beads (Figure 8B) at 1648 cm^−1^ overlapping with the vibration band at 1646 cm^−1^ obtained in the spectrum without cross-linking, and the cross-linking reaction is evidenced by the small increased intensity of the stretch band. A small increase in the 2875 cm^−1^ band corresponding to the C-H crosslinked bond is overlapped with the -CH_2_- groups in the GA structure. In general, the spectrum showed an increase in intensity in the bands attributed to the bands generated by the various CS bonds (NH; OH; CO) overlapped with the GA structure’s bonding groups [31].

Finally, the X-ray diffraction pattern of the non-reticulated CS beads (Figure 9) presented three peaks at 2θ values of 10°, 20°, and 41°, revealing the existence of amorphous regions and high crystallinity. This result is consistent with previous studies reported in the literature [23,48,49,50,51].

The high crystallinity is due to the arranged saccharide structure since the hydroxyl and amino groups could form strong intermolecular and intramolecular hydrogen bonds [52].

Furthermore, the structure of CS molecules has a certain regularity. As a result, CS molecules form crystalline regions [51,53]. In crosslinking CS with GA, it was observed that the peak seen in the CS spectrum at 2θ = 10.3° disappeared, and the peak at 2θ = 20° considerably decreased its intensity. According to previous studies [52,54,55], the GA-crosslinked CS beads’ crystallinity decreases concerning the unmodified CS. This crystallinity reduction could be attributed to the strong hydrogen bond’s deformation in the original CS due to hydroxyl and amino groups’ substitution, resulting in an amorphous structure [53]. The beads’ crystallinity index (Xc) showed values of 25% for the non-crosslinked CS and 16% for the GA-crosslinked CS. This confirms that the addition of a component, such as GA, decreases the crystallinity of CS, since it inhibits the folding of the polymer chains responsible for producing crystalline regions in the polymeric network [51].

### 3.5. Kinetic Model Analysis

To determine the kinetic model that best fits the experimental data and describes the adsorption mechanism, the non-linear models of pseudo-first order (PFO), pseudo-second order (PSO), Elovich, and intraparticle diffusion were analyzed. Figure 10 shows the fit of the kinetic models. The values obtained from the maximum adsorption capacities, the speed constants, and the regression coefficient for each model are presented in Table 5.

The experimental results showed that the adsorption equilibrium time is reached at 465 min, which corresponded to an 87 % removal efficiency and an adsorption capacity of 1.16 mg/g. Of the models applied in the kinetics, the Elovich model better fit because of the highest determination coefficient and represents the adsorption kinetics of the RB4 dye on the GA-crosslinked CS beads. The constant α (mg g^−1^ min^−1^) corresponds to the initial adsorption rate, and β is related to the degree of coverage and activation energy of the surface [41]. Elovich’s model suggests that the chemisorption mechanism controlled the dye’s adsorption process on the beads and the adsorption rate decreased with time due to more generous coverage on the surface [41,42,43,56,57].

### 3.6. Adsorption Mechanism of RB4 Dye

The adsorption mechanism of the RB4 dye on the surface of the GA-crosslinked CS beads is shown in Figure 11. The electrostatic attractions between the negatively charged sulfonate groups (SO_3_^−^) of the RB4 dye with the amino group (NH_3_^+^) and the positively charged hydroxyl group (HO^+^–H) available on the surface GA-crosslinked CS beads are favored when the total "charge" of CS adsorbent is highly cationic, due to the stronger protonation of functional groups at acid pH [58]. Other interactions that take place, to a lesser extent, in the adsorption mechanism are the formation of dipole–dipole bonds, the Yoshida bond, and n–pi interactions [46,59,60,61,62].

### 3.7. Isothermal Adsorption Equilibrium

Figure 12 shows the isothermal models (Langmuir, Freundlich, Temkin) used to analyze the RB4 dye adsorption’s equilibrium characteristics. The parameters obtained for each model are presented in Table 6.

The Freundlich isotherm model provided a better fit with the experimental equilibrium data. This indicated that the system’s adsorption process is not limited by generating the monolayer of molecules of the RB4 dye. Instead, there is a heterogeneous distribution of the active sites on the GA-crosslinked CS beads’ surface. The constant *K_F_* of the model indicated an adsorption capacity of 0.933 when equilibrium was reached. In this sense, the most attractive junction sites occupied the first spaces, and the energy of adsorption decreased with the increasing degree of occupancy of the active adsorption sites [42,63,64]. The parameter n, known as the adsorption intensity, indicated the system’s adsorption process’s favorability. As 1 < n (5.121) < 10, it was determined that the GA-crosslinked CS beads exhibited good adsorption characteristics [65,66]. Table 7 shows that the chemisorption mechanism can control the adsorption processes of RB4 against various adsorbents. Likewise, the distribution of the molecules between the liquid and solid phase depends on the adsorbent’s affinities with RB4.

The sorption capacities obtained by various adsorbents in the removal of RB4 are compared in Table 8. In the first place, the adsorbents described exhibit different values in their maximum adsorption capacities due to the treatment’s effectiveness depending on the experimental conditions, such as the dose of the adsorbent, the initial concentration of the adsorbate, and the pH, among others. The high doses of the adsorbents favor the overlap of the available surface of the material. Thus, the total effective surface area can decrease, thus decreasing the adsorption [67]. On the other hand, the high initial concentrations of RB4 favor the adsorption capacity since the material’s active sites are occupied to a greater extent until reaching complete saturation [68]. Likewise, it is evidenced that the adsorption capacity of RB4, in general, is more effective when the pH takes acid values. In this study, it is reported that the maximum adsorption capacity was 1.56. It is possible that the adsorbent still has active sites for adsorption.

## 4. Conclusions

The present study experimentally demonstrated that RSM designs allow locating the most suitable conditions where the response is maximized. The mathematical models obtained in the RSM designs provided good fits with the experimental data at a 95% confidence level. A maximum predicted value of 60.65% was obtained for the experimental design, where the swelling degree was applied as a response variable. Likewise, it was found that the GA cross-linked CS beads increased their degree of swelling at acidic pH due to the protonation of the amino groups on the beads, which acted as repulsive forces between the polymer chains. The complete elimination of RB4 was achieved in the experimental design’s elimination efficiency.

The crosslinking reaction produced between CS and GA was demonstrated through SEM images, which showed the crosslinking’s rough surface characteristic. Crystallinity and chemical structure changes were confirmed by XRD and FTIR spectra. The existence of amorphous and high crystallinity regions of the CS beads was identified at values of 2θ = 10°, 20°, and 41°. When crosslinking with GA occurred, a loss of crystallinity occurred, eliminating the peak at 10° and decreasing the peak at 20°. The crystallinity index also confirmed the decrease in the non-crosslinking CS’s crystallinity from 25% to 16% with the crosslinking agent’s addition. In the FTIR, beads’ crosslinking was supported by an imine bond stretching band (C=N) at 1648 cm^−1^ that was found to overlap with the vibration band of 1646 cm^−1^ the functional group C=O. The adsorption kinetics suggested that the interaction mechanism was controlled by chemisorption. The equilibrium data indicated that the RB4 molecules are heterogeneously distributed in the active sites on the CS/GA beads’ surface. Finally, the GA crosslinked beads presented in this work can act as an ecological adsorbent, capable of efficiently removing anionic dyes in water bodies.

## Figures and Tables

**Figure 1 life-11-00085-f001:**
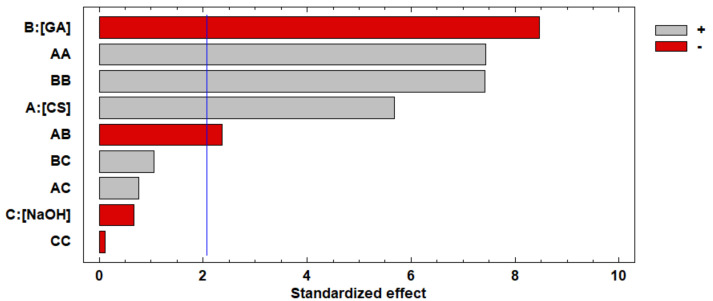
Pareto chart of standardized effects for response variable swelling degree.

**Figure 2 life-11-00085-f002:**
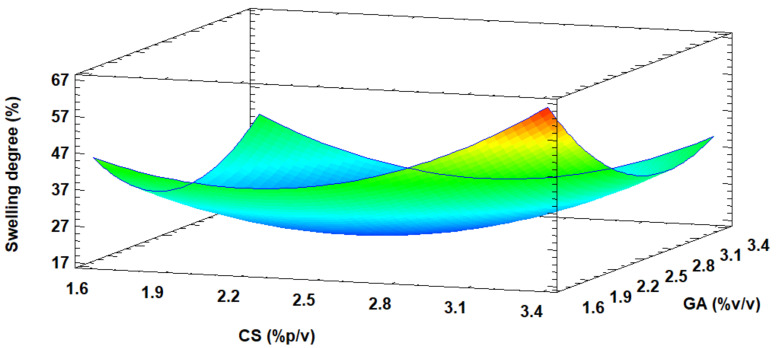
Three-dimensional (3D) response surface plot for optimizing the swelling degree of GA-crosslinked CS beads. NaOH = 1.5 M.

**Figure 3 life-11-00085-f003:**
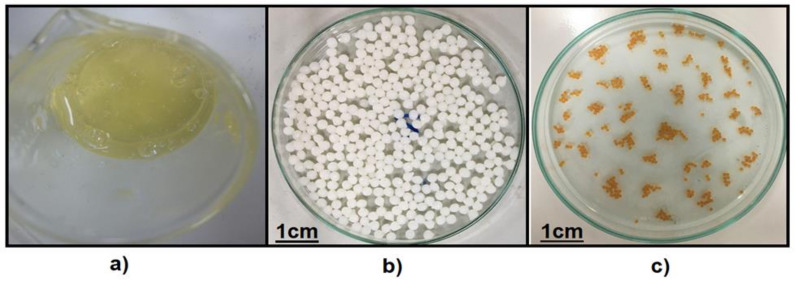
The appearance of CS in the process of obtaining the beads; (**a**) CS–CH_3_COOH solution, (**b**) CS beads precipitated in NaOH_(aq)_; (**c**) final product of the beads cross-linked with GA.

**Figure 4 life-11-00085-f004:**
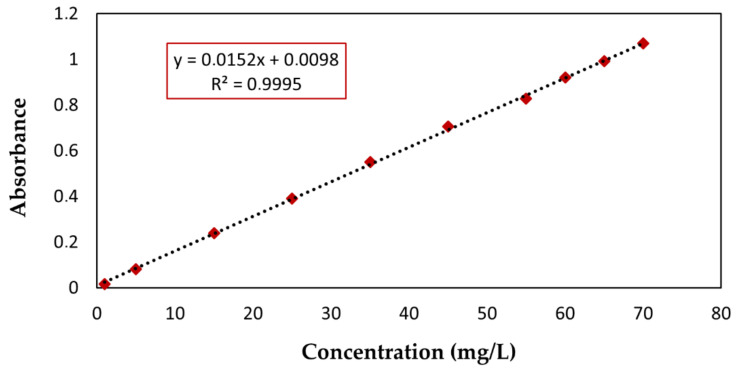
Calibration curve for RB4 dye measured at 599 nm. Concentration range of 1 mg/L to 70 mg/L.

**Figure 5 life-11-00085-f005:**
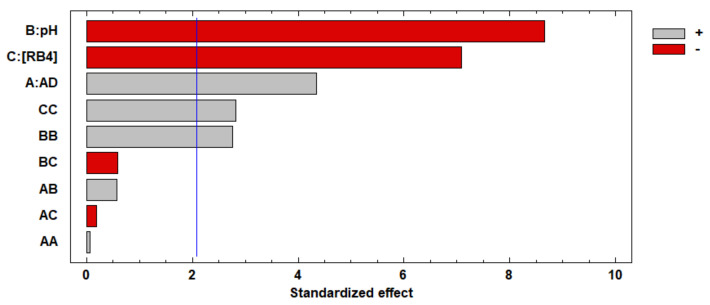
Pareto chart of standardized effects for response variable RB4 dye removal efficiency. AD: adsorbent dose.

**Figure 6 life-11-00085-f006:**
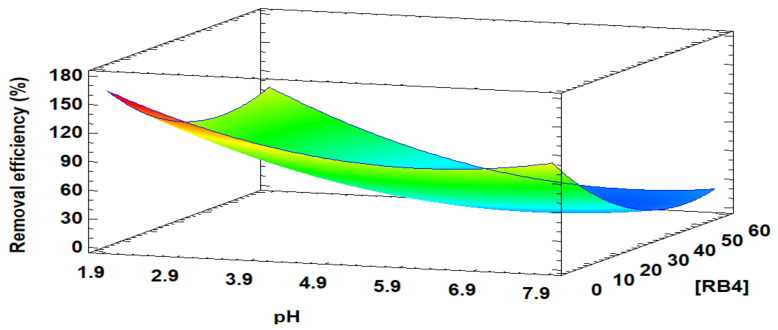
The 3D response surface plot for optimization of RB4 dye removal efficiency. Adsorbent dose: 0.35 g/25 mL.

**Figure 7 life-11-00085-f007:**
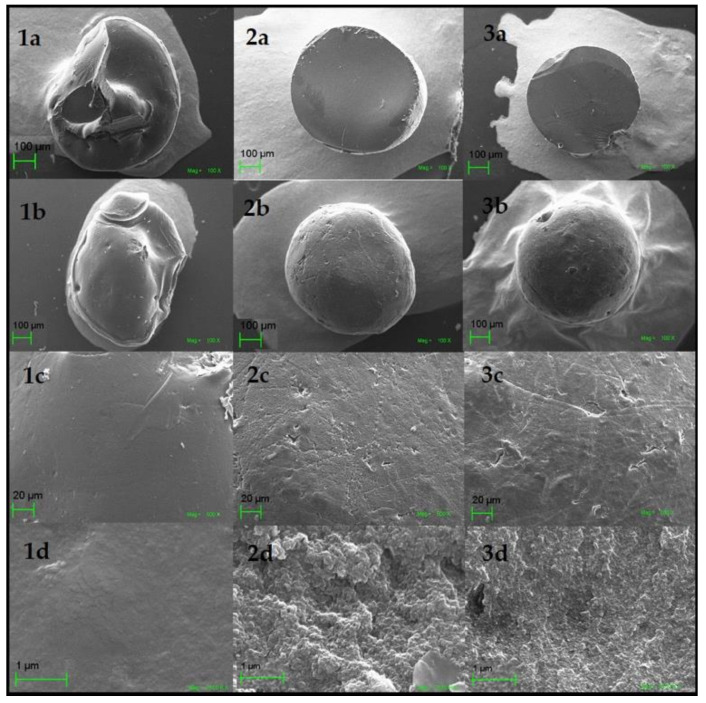
SEM images. Morphology of non-crosslinked CS beads, cross-section: (**1a**) at 100×, surface area: (**1b**) at 100×, (**1c**) at 500×, (**1d**) at 25,000×; GA-crosslinked CS beads, cross-section (**2a**) at 100×, surface area: (**2b**) at 100×, (**2c**) at 500×, (**2d**) at 25,000×; GA-crosslinked CS beads after adsorption of the RB4 dye, cross-section (**3a**) at 100×, surface area: (**3b**) at 100×, (**3c**) at 500×, (**3d**) at 25,000×.

**Figure 8 life-11-00085-f008:**
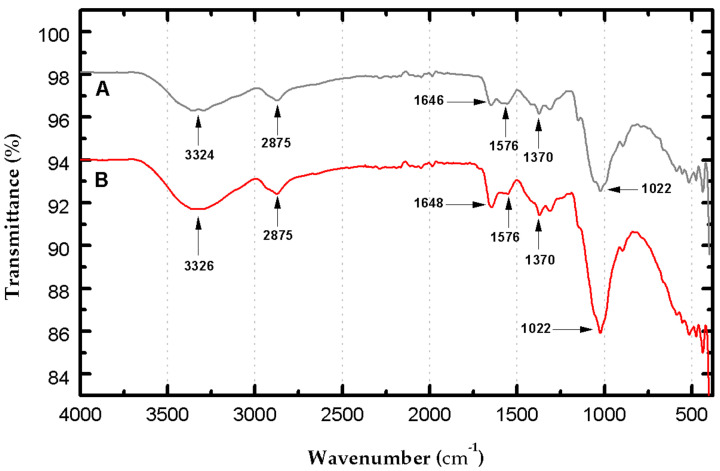
FT-IR spectrum of non-crosslinked CS beads (**A**) and GA-crosslinked CS beads (**B**).

**Figure 9 life-11-00085-f009:**
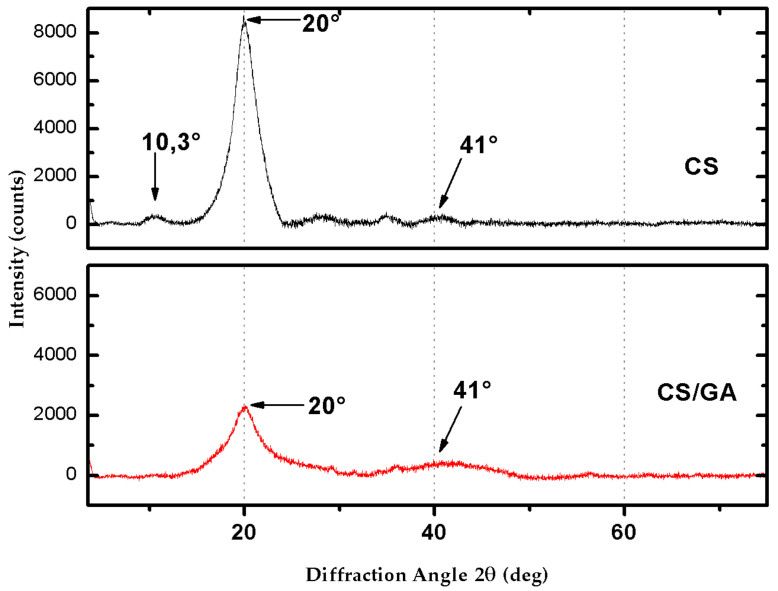
X-ray diffractogram of non-crosslinked CS beads (upper-black line) and GA-crosslinked CS beads (lower-red line).

**Figure 10 life-11-00085-f010:**
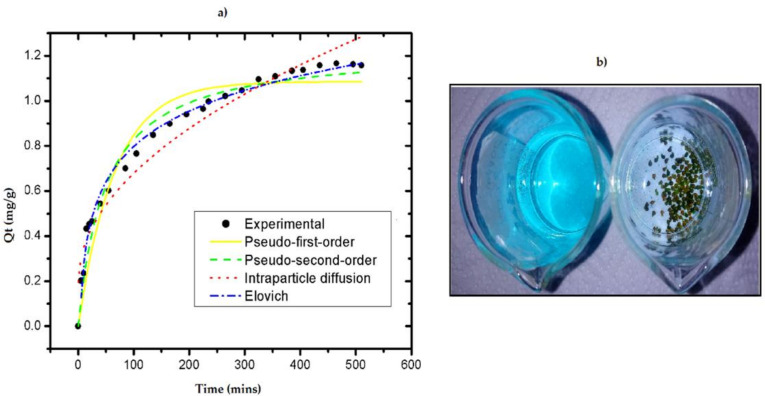
(**a**) Adsorption kinetics adjusted to different models for RB4 dye; (**b**) aqueous solution of RB4 dye before and after the adsorption process with GA-crosslinked CS beads. (Initial concentration of RB4 dye = 35 mg/L; pH = 3.0; adsorbent dose = 0.4 g; solution volume = 25 mL; constant stirring = 400 rpm; temperature = 25 °C).

**Figure 11 life-11-00085-f011:**
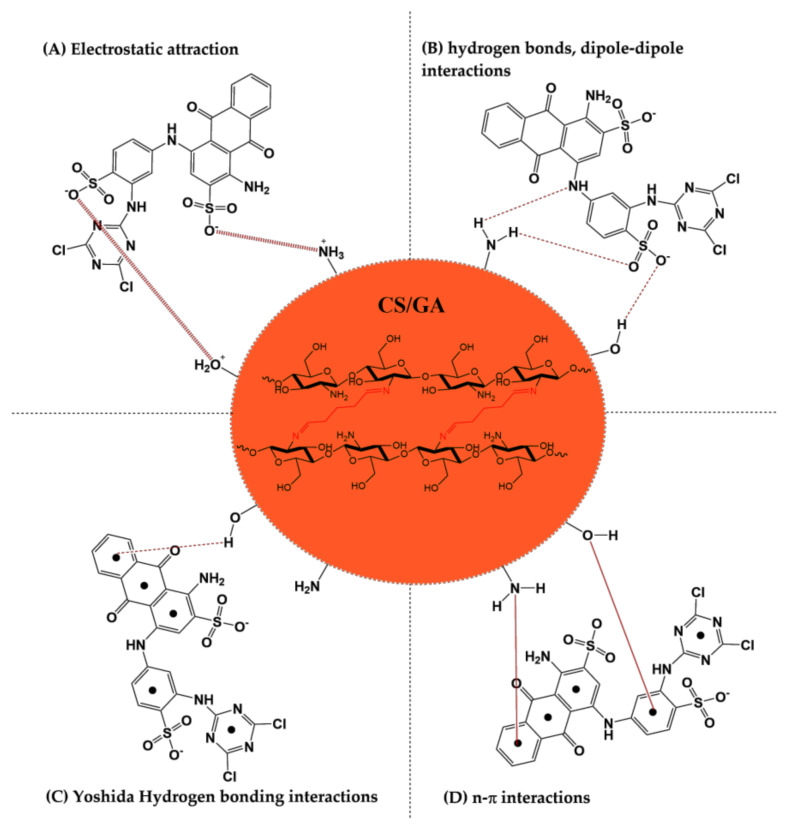
Illustration of the possible interactions between GA-crosslinked CS beads (CS/GA) and RB4 dye: (**A**) electrostatic attraction, (**B**) hydrogen bonds dipole-dipole interactions, (**C**) Yoshida hydrogen bonding interactions, (**D**) n– π interactions.

**Figure 12 life-11-00085-f012:**
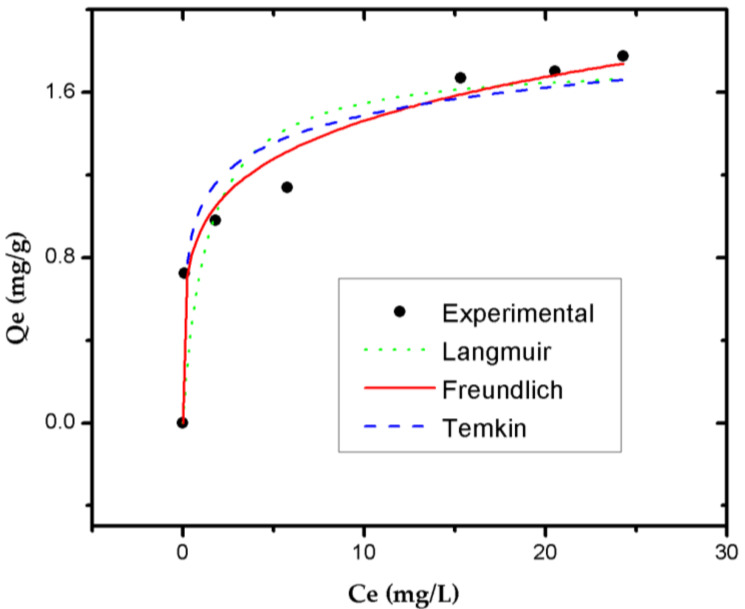
Equilibrium adsorption adjusted to different isothermal models for RB4 dye adsorption on GA-crosslinked CS beads (Initial concentration of RB4 dye = 35 mg/L; pH = 3.0; adsorbent dose = 0.4 g/25 mL; constant stirring = 400 rpm; temperature = 25 °C).

**Table 1 life-11-00085-t001:** Experimental design matrix and response results for glutaraldehyde (GA)-crosslinked chitosan (CS) beads preparation.

Factors				Response	
Run	CS (% *w/v*)	GA (% *v/v*)	NaOH (M)	Swelling Degree (%)	
**1**	2.5	2.5	1.5	19.23	18.83
**2**	3.0	3.0	2.0	25.73	25.00
**3**	2.5	3.3	1.5	22.50	21.34
**4**	3.0	3.0	1.0	23.58	23.06
**5**	2.0	2.0	1.0	30.07	29.82
**6**	2.0	3.0	1.0	24.15	26.15
**7**	2.5	2.5	2.3	17.57	17.23
**8**	2.5	2.5	1.5	18.06	19.51
**9**	3.0	2.0	1.0	33.99	35.01
**10**	1.7	2.5	1.5	23.15	21.80
**11**	2.5	2.5	0.7	18.92	19.32
**12**	2.0	3.0	2.0	24.88	25.12
**13**	2.0	2.0	2.0	28.47	26.11
**14**	3.0	2.0	2.0	33.16	34.03
**15**	2.5	1.7	1.5	40.54	40.12
**16**	3.3	2.5	1.5	39.71	39.88

**Table 2 life-11-00085-t002:** ANOVA results for GA-crosslinked CS beads preparation experiments.

Factors	Sum of Squares	df ^a^	Mean Square	F	Value-*p* ^b^
A: (CS)	215.986	1	215.986	30.47	0.0000
B: (GA)	481.445	1	481.445	67.93	0.0000
C: (NaOH)	2.88398	1	2.88398	0.410	0.5304
AA	373.589	1	373.589	52.71	0.0000
AB	38.7195	1	38.7195	5.460	0.0294
AC	3.65766	1	3.65766	0.520	0.4804
BB	373.001	1	373.001	52.63	0.0000
BC	7.77016	1	7.77016	1.100	0.3070
CC	0.07039	1	0.07039	0.010	0.9216
Blocks	0.49252	1	0.49252	0.070	0.7946
Total error	148.836	21	7.08744		
Total (corrected)	1608.0	31			
R^2^	90.74				
Adj.-R^2^	86.34				
Adequate Precision	14.11				

^a^ degree of freedom, ^b^ considered significant when *p* < 0.050.

**Table 3 life-11-00085-t003:** Experimental design matrix and response results for RB4 dye adsorption on GA-crosslinked CS beads.

Factors				Response	
Run	AD (g)	pH	[RB4] (mg/L)	Removal Efficiency (%)	
				1	2
**1**	0.35	4.5	55	31.81	32.71
**2**	0.50	3.0	45	74.05	73.39
**3**	0.50	6.0	15	90.00	79.59
**4**	0.35	4.5	5,0	78.22	70.99
**5**	0.20	6.0	15	42.84	47.98
**6**	0.20	3.0	45	46.06	42.62
**7**	0.35	2.0	30	98.34	96.19
**8**	0.35	4.5	30	28.87	28.48
**9**	0.60	4.5	30	50.06	49.72
**10**	0.20	6.0	45	14.75	13.21
**11**	0.35	7.0	30	7.465	9.566
**12**	0.50	3.0	15	100.0	99.72
**13**	0.35	4.5	30	53.76	43.41
**14**	0.20	3.0	15	97.49	94.77
**15**	0.10	4.5	30	9.525	9.853
**16**	0.50	6.0	45	21.11	24.46

AD: Adsorbent Dose; [RB4]: concentration of RB4.

**Table 4 life-11-00085-t004:** ANOVA results for RB4 dye adsorption experiments on GA-crosslinked CS beads.

Factors	Sum of Squares	Df ^a^	Mean Square	F	Value-*p* ^b^
A: Adsorbent dose	3247.78	1	3247.78	18.92	0.0003
B: pH	12,860.1	1	12,860.1	74.93	0.0000
C: (RB4)	8618.51	1	8618.51	50.22	0.0000
AA	0.878497	1	0.878497	0.01	0.9436
AB	56.9199	1	56.9199	0.33	0.5708
AC	6.09537	1	6.09537	0.04	0.8523
BB	1302.50	1	1302.50	7.59	0.0119
BC	60.1373	1	60.1373	0.35	0.5602
CC	1362.49	1	1362.49	7.94	0.0103
Blocks	23.9524	1	23.9524	0.14	
Total error	3604.16	21	171.627		
Total	30,923.8	31			
R^2^	88.35				
Adj.-R^2^	82.80				
Adequate Precision	14.03				

^a^ degree of freedom, ^b^ considered significant when *p* < 0.050.

**Table 5 life-11-00085-t005:** Kinetics model parameters for RB4 dye adsorption on GA-crosslinked CS beads.

Model	Parameters		Adj.-R^2^
PFO	*q_e_* (mg/g)	*K_1_* (min^−1^)	0.9287
	1.08558	0.01522	
PSO	*q_e_* (mg/g)	*K_2_* (g∗(mg∗min)^−1^)	0.9720
	1.23559	0.01646	
Intraparticle diffusion	*K_i_* (mg/g∗min^−0.5^)	*C* (mg/g)	0.9484
	0.04812	0.19723	
Elovich	α (mg/g∗min)	β (mg/g)	0.9893
	0.07708	4.42261	

PFO: pseudo-first order; PSO: pseudo-second order.

**Table 6 life-11-00085-t006:** Equilibrium models and their calculated parameters for RB4 dye adsorption on GA-crosslinked CS beads.

Model	Parameters		Adj.-R^2^
Langmuir	*q_m_* (mg/g)	*K_L_* (L/mg)	0.7736
	1.756	0.747	
Freundlich	*n*	*K_F_* [(mg/g)(mg∗L^−1/n^]	0.9708
	5.121	0.933	
Temkin	*K_t_* (L/mg)	*bt*	0.8597
	223.668	12,827.383	

**Table 7 life-11-00085-t007:** Comparison of the kinetic and isothermal models in the adsorption of RB4 in different adsorbents.

Adsorbent	Kinetic	Isotherm	Ref.
CS/activated charcoal	Pseudo-second order	Langmuir	[67]
Seeds of *Moringa oleifera*@MnFe_2_O_4_	Pseudo-second order	Freundlich	[68]
Microcrystalline cellulose–epichlorohydrin	Pseudo-second order	Langmuir	[69]
Polymetallic nanoparticles	Pseudo-second order	Langmuir	[70]
guar gum and silica nanocomposite	Pseudo-second order and intraparticle diffusion	Langmuir	[71]
Pecan nutshells	Pseudo-second order	Langmuir	[72]
Rice bran/Fe_3_O_4_	Pseudo-second order	Langmuir	[73]
CS/hexadecylamine	Pseudo-second order	Freundlich	[20]
CS/hexadecylamine /3-aminopropyl triethoxysilane	Pseudo-second order	Freundlich	[74]
CS/3-aminopropyl triethoxysilane	Pseudo-second order	Freundlich	[35]
GA-crosslinked CS beads	Elovich	Freundlich	This study

**Table 8 life-11-00085-t008:** Comparison of the adsorption capacities of various materials for the removal of RB4.

Adsorbent	Adsorbent Concentration (g/L)	Adsorbate Concentration (mg/L)	pH	Maximum Adsorption Capacity (mg/g)	Ref.
Coconut shells (biomass)	10	1.274	5	0.0064	[75]
Cauliflower cores (biomass)	10	1.274	5	0.032	[75]
CS/activated Charcoal	0.1/0.1	100	7	250	[67]
Microcrystalline cellulose–epichlorohydrin	1.0	200	3	70	[69]
Dry cells of *Rhizopus oryzae*	4.0	100	3	24	[76]
Polymetallic nanoparticles	0.4	200	8.5	345	[70]
guar gum and silica nanocomposite	0.03/0.025	200	2	579	[71]
Pecan nut shells	10	1000	6.5	5	[72]
La(III) supported carboxymethylcellulose-clay	0.1	50	3	43.65	[77]
Rice bran/Fe_3_O_4_	1.5	200	2	185.2	[73]
Rice bran/SnO_2_/Fe_3_O_4_	1.5	200	2	218.8	[78]
Nickel-metal hydride spent batteries	0.80	200	3	331	[79]
Seeds of moringa oleifera@MnFe_2_O_4_	0.1/0.05	50	3	32.45	[68]
CS/hexadecylamine	0.2/0.2	500	4	454	[22]
CS/3-aminopropyl triethoxysilane Beads	0.2/0.2	500	4	433.7	[35]
CS/hexadecylamine/3-aminopropyl triethoxysilane	0.2/0.2	500	4	468.8	[74]
Extracellular polymeric substances	0.1/0.05	50	2	42.93	[80]
CS 10B (100% deacetylated chitin)	0.05/0.025	63.74	4	54,01	[81]
CS/hydroxyapatite	0.03	950	4	118.4	[82]
GA-crosslinked CS beads	0.4/0.025	55	3	1.56	This study

## Data Availability

Not applicable

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
