# Peer review of "Optimization of Chitosan Glutaraldehyde-Crosslinked Beads for Reactive Blue 4 Anionic Dye Removal Using a Surface Response Methodology"

_life, 2021, doi:10.3390/life11020085_

Round 1

Reviewer 1 Report

This is a very interesting paper and I recommand it for publication in Life. Only some typing errors should be corrected. The paper is very clear and it is very well written.

  • Fig. 9: Diffraction Angle /In the text it is correctly written diffraction/
  • lines 411 and 412 should become: dipole-dipole bonds, the Yoshida bond and n-pi interactions [47, 60-63].
  • In Fig 12: (B) hydrogen bonds, dipole-dipole interactions
  • line 486: Names in minuscules
  • Please do check the text of the references

Author Response

Reviewer 1

Fig. 9: Diffraction Angle /In the text it is correctly written diffraction/

R// We appreciate the reviewer's comment. Figure  9 was modified.

Lines 411 and 412 should become: dipole-dipole bonds, the Yoshida bond, and n-pi interactions [47, 60-63].

R// We appreciate the reviewer's comment. Lines 411 and 412 were modified accordingly.

In Fig 12: (B) hydrogen bonds, dipole-dipole interactions

R// We appreciate the reviewer's comment. Figure 12 was modified.

Line 486: Names in minuscules. Please do check the text of the references.

R// We appreciate the reviewer's comment. Names were modified correctly, and the text of the references was checked.

Reviewer 2 Report

The paper describes the modification of chitosan with glutharaldehyde and the potential use of the obtained adsorbent for dyes removal from aqueous solutions.

Unfortunately, the research subject is not innovative. There are many reports in the literature about the use of this adsorbent for dyes removal. However, the paper does not compare the sorption capacity of the investigated adsorbent with other adsorbents in relation to the dye (RB4). Such a comparison allows an objective assessment of the sorption properties of a given adsorbent. In this section of the paper, as in the section on kinetics, there is no comparison with available literature data. This is the main disadvantage of this work. 

It is also necessary to assess the sorption properties in the presence of surfactants or electrolytes present in dye baths containing reactive dyes such as RB4.

The possibility of dye desorption from the adsorbent phase was not investigated.

The paper does not include kinetic and isothermal equations (which can be added in the Supplementary Materials). 

The above remarks lead me to the following evaluation of the paper: it requires major corrections.

Author Response

Reviewer 2

Unfortunately, the research subject is not innovative. There are many reports in the literature about the use of this adsorbent for dyes removal. However, the paper does not compare the sorption capacity of the investigated adsorbent with other adsorbents in relation to the dye (RB4). Such a comparison allows an objective assessment of the sorption properties of a given adsorbent. In this section of the paper, as in the section on kinetics, there is no comparison with available literature data. This is the main disadvantage of this work.

 R// We appreciate the reviewer's comment. The innovative aspect of this study is the optimization of the variables that influence the adsorption process of modified chitosan beads with response surface methodology (RSM), which determines and quantifies the factors that could influence the variable of interest. The isothermal equilibria discussion in this study against other findings is shown between lines 430 to 432 and Table 7. The distribution of the molecules between the liquid and solid phase depends on the adsorbent's affinities with RB4. Therefore, the data is expected to respond to the Langmuir or Freundlich model. We compared other adsorbents and our system for the RB4 adsorption capacity and some discussion in table 8 and lines 432 to 448.

It is also necessary to assess the sorption properties in the presence of surfactants or electrolytes present in dye baths containing reactive dyes such as RB4.

R// We appreciate the reviewer's suggestion. However, surfactants or electrolytes were not assessed. Our experimental design did not include the ionic strength effect of surfactants present. Several publications reviewed did not include the ionic strength effect in the adsorption capacity of the adsorbents. For that reason, we did not include in the experimental design the ionic strength modification as a variable. In the future, these variables might be included to see the effect.

  1. Hong, G.-B.; Wang, Y.-K. Synthesis of low-cost adsorbent from rice bran for the removal of reactive dye based on the response surface methodology. Appl. Surf. Sci. 2017, 423, 800–809, doi:https://doi.org/10.1016/j.apsusc.2017.06.264.
  2. Vakili, M.; Rafatullah, M.; Ibrahim, M.H.; Abdullah, A.Z.; Gholami, Z.; Salamatinia, B. Enhancing reactive blue 4 adsorption through chemical modification of chitosan with hexadecylamine and 3-aminopropyl triethoxysilane. J. Water Process Eng. 2017, 15, 49–54.
  3. Vakili, M.; Rafatullah, M.; Salamatinia, B.; Ibrahim, M.H.; Abdullah, A.Z. Elimination of reactive blue 4 from aqueous solutions using 3-aminopropyl triethoxysilane modified chitosan beads. Carbohydr. Polym. 2015, 132, 89–96.
  4. Vakili, M.; Rafatullah, M.; Ibrahim, M.H.; Abdullah, A.Z.; Salamatinia, B.; Gholami, Z. for improved reactive blue 4 adsorptioChitosan hydrogel beads impregnated with hexadecylaminen. Carbohydr. Polym. 2016, 137, 139–146
  5. Karaer, H.; Kaya, I. Synthesis, characterization of magnetic chitosan/active charcoal composite and using at the adsorption of methylene blue and reactive blue4. Microporous Mesoporous Mater. 2016, 232, 26–38.

Ma, C.M.; Hong, G.B.; Wang, Y.K. Performance evaluation and optimization of dyes removal using Rice bran-based magnetic composite adsorbent. Materials (Basel). 2020, 13, 2764.

The possibility of dye desorption from the adsorbent phase was not investigated.

R// We appreciate the reviewer's comment. However, the dye desorption will be investigated in an upcoming publication with other systems as well. In the present study, dye desorption was not the main focus of the work.

The paper does not include kinetic and isothermal equations (which can be added in the Supplementary Materials). 

R// We appreciate the reviewer's comment. The kinetic (Table S1) and isothermal (Table S2) models were added to the supporting information.

Reviewer 3 Report

The study shown in the paper 'Optimization of Chitosan Glutaraldehyde-Crosslinked Beads for Reactive Blue 4 Anionic Dye removal using a Surface Response Methodology' is interesting, although the maximum adsorption capacity is not too high. From my point of view this paper can be published after some revisions in order to make some passages clearer:

Line 25: the response surface methodology (RMS)…., Line 80: response surface methodology (RSM)…., what kind of abbreviation should be used?

The appropriate value of adequate precision should be mentioned in the article. It is important for the adjusted quadratic model.

Line 132: Why is the dye concentration prepared in a range of 5 mg/L to 55 mg/L?

For practical industrial applications, adsorbent must also demonstrate the stability of cycles. The stability of adsorbent needs to be supplemented.

After the first appearance of the abbreviation, the abbreviation should always be used in the rest of the manuscript instead of the complete term. (ex: glutaraldehyde, chitosan, scanning electron microscopy,….)

The findings reported in isothermal adsorption equilibrium need to be discussed with results from other studies. This will allow for a better understanding of the significance of the findings.

Author Response

Reviewer 3

Line 25: the response surface methodology (RMS)…., Line 80: response surface methodology (RSM)…., what kind of abbreviation should be used?

R// We appreciate the reviewer's comment. Line 80 was changed to the correct abbreviation RSM.

The appropriate value of adequate precision should be mentioned in the article. It is important for the adjusted quadratic model.

R// We appreciate the reviewer's comment. The value of adequate precision was added in Table 2 and Table 4 for adjusted quadratic models.

Line 132: Why is the dye concentration prepared in a range of 5 mg/L to 55 mg/L?

R // We appreciate the reviewer's comment. We could define the dye concentration range for the adsorption experiments from preliminary studies and the literature. To define the range of the design variables, the influencing variables that would remain constant were established. For example, the contact time of each run was established at 48 hours. Therefore, to determine the dye concentration range from preliminary tests, it was taken into account that, in 48 hours, the adsorption equilibrium would be ensured up to 55 mg/L. However, the mathematical model obtained for the RSM under this experimental strategy showed a high predictive quality, which led to finding the most suitable operating conditions for the RB4 dye removal process.

For practical industrial applications, adsorbent must also demonstrate the stability of cycles. The stability of the adsorbent needs to be supplemented.

R// We appreciate the reviewer's comment. However, the dye desorption will be investigated in an upcoming publication with other systems as well. In the present study, dye desorption was not the main focus of the work.

After the first appearance of the abbreviation, the abbreviation should always be used in the rest of the manuscript instead of the complete term. (ex: glutaraldehyde, chitosan, scanning electron microscopy,….)

R// We appreciate the reviewer's comment. Their corresponding abbreviations have replaced all the terms.

The findings reported in isothermal adsorption equilibrium need to be discussed with results from other studies. This will allow for a better understanding of the significance of the findings.

R// We appreciate the reviewer's comment. The isothermal equilibria discussion in this study against other findings is shown between lines 430 to 432 and Table 7. The molecules' distribution between the liquid and solid phase depends on the adsorbent's affinities with RB4. Therefore, the data is expected to respond to the Langmuir or Freundlich model. We compared other adsorbents and our system for the RB4 adsorption capacity and some discussion in table 8 and lines 432 to 448.

Round 2

Reviewer 2 Report

The authors of the paper have addressed the comments made and the paper maybe published in its current form.

Reviewer 3 Report

I recommended that the manuscript could be accepted for publication.